# Advances in DNA/RNA Sequencing and Their Applications in Acute Myeloid Leukemia (AML)

**DOI:** 10.3390/ijms26010071

**Published:** 2024-12-25

**Authors:** Fatimah Ahmed, Jiang Zhong

**Affiliations:** Department of Basic Sciences, Loma Linda University School of Medicine, Loma Linda, CA 92350, USA; fmrittika@gmail.com

**Keywords:** acute myeloid leukemia (AML), whole-genome sequencing (WGS), whole-exome sequencing (WES), RNA sequencing (RNA-Seq), precision medicine

## Abstract

Acute myeloid leukemia (AML) is an aggressive malignancy that poses significant challenges due to high rates of relapse and resistance to treatment, particularly in older populations. While therapeutic advances have been made, survival outcomes remain suboptimal. The evolution of DNA and RNA sequencing technologies, including whole-genome sequencing (WGS), whole-exome sequencing (WES), and RNA sequencing (RNA-Seq), has significantly enhanced our understanding of AML at the molecular level. These technologies have led to the discovery of driver mutations and transcriptomic alterations critical for improving diagnosis, prognosis, and personalized therapy development. Furthermore, single-cell RNA sequencing (scRNA-Seq) has uncovered rare subpopulations of leukemia stem cells (LSCs) contributing to disease progression and relapse. However, widespread clinical integration of these tools remains limited by costs, data complexity, and ethical challenges. This review explores recent advancements in DNA/RNA sequencing in AML and highlights both the potential and limitations of these techniques in clinical practice.

## 1. Introduction

Acute myeloid leukemia (AML) is an aggressive hematologic malignancy characterized by the clonal expansion of myeloid progenitor cells in the bone marrow and blood, leading to impaired normal hematopoiesis and life-threatening complications [1,2,3]. In 2024, approximately 20,800 new AML cases were reported in the United States, representing about 1% of all cancer diagnoses [4]. The 5-year relative survival rate stands at 31.9% [4], with younger patients achieving survival rates of up to 50%, while individuals over 60 experience survival rates below 10% [5]. Although many patients achieve complete remission (CR) after initial chemotherapy, relapse occurs in the majority, often within the first three years [6], particularly in older adult populations [7]. Despite advancements in chemotherapy, targeted therapies, and stem cell transplantation, survival outcomes remain suboptimal for patients, particularly those harboring high-risk cytogenetic and molecular profiles [2,8,9]. These challenges highlight the ongoing need for more effective and personalized therapeutic strategies.

In recent years, the molecular complexity of AML has been increasingly elucidated through advances in high-throughput sequencing technologies, particularly next-generation sequencing (NGS) [10,11,12]. NGS approaches, including whole-genome sequencing (WGS), whole-exome sequencing (WES), and RNA sequencing (RNA-Seq), have greatly enhanced our understanding of the genetic and transcriptomic landscape of AML. These technologies have enabled the identification of key driver mutations in genes involved in epigenetic regulation, transcription, and signal transduction, such as NPM1, FLT3, DNMT3A, and IDH1/2, which have significant prognostic and therapeutic implications [10,12]. Furthermore, single-cell RNA sequencing (scRNA-Seq) has emerged as a powerful tool for uncovering intratumoral heterogeneity and rare subpopulations of leukemia stem cells, which may contribute to disease relapse and resistance to treatment [13,14]. The integration of genomic data into clinical practice is now paving the way for more personalized treatment approaches, although challenges such as data complexity, cost, and ethical concerns persist. This review will discuss the recent advancements in DNA and RNA sequencing technologies and their applications in AML, as well as the challenges and future directions in this rapidly evolving field.

## 2. Advances in DNA Sequencing

### 2.1. Whole-Genome Sequencing (WGS)

Whole-genome sequencing (WGS) captures the complete genetic blueprint, including both coding and non-coding regions, making it a critical tool in genomic research. Since the first successful sequencing of the human genome in 2003, WGS has played an essential role in understanding complex diseases, including cancer [15,16]. WGS offers a comprehensive snapshot of the genome, enabling the identification of rare variants, copy number variations (CNVs), and structural alterations that are often implicated in diseases such as acute myeloid leukemia (AML) [17].

DNA breakpoint cloning has become a crucial technique for monitoring minimal residual disease (MRD) in both chronic and acute myeloid leukemia (CML and AML) [18]. It provides insights into specific genetic breakpoints associated with chromosomal translocations and mutations. WGS enhances MRD detection by offering a more detailed analysis of genetic variations, including structural changes that traditional methods often miss [19]. Recent advancements in NGS-based breakpoint cloning have notably improved MRD detection, demonstrating superior sensitivity compared to traditional techniques like polymerase chain reaction (PCR) [20]. These breakthroughs are essential for identifying clonal populations and variants associated with disease relapse, thereby supporting the development of more personalized treatment strategies.

Among structural variations, balanced translocations and inversions are frequently associated with AML and play a critical role in its pathogenesis [21]. These chromosomal abnormalities can lead to the formation of fusion genes that contribute to leukemogenesis [22]. WGS has emerged as a powerful tool for detecting these abnormalities with high accuracy, surpassing traditional cytogenetic techniques such as fluorescence in situ hybridization (FISH) and karyotyping, which are limited in resolution and scope [23]. By enabling comprehensive genomic analysis and identifying submicroscopic variations, WGS holds the potential to enhance diagnostic precision and efficiency. Integrating WGS into diagnostic workflows could provide direct comparisons with standard methods, improving patient stratification and treatment outcomes in AML.

Recent research underscores the utility of WGS in uncovering novel mutations and structural variations in AML. A 2023 study demonstrated how WGS identified critical translocations and mutations in DNMT3A, NPM1, and FLT3, which are central to AML pathogenesis [3,17]. Furthermore, WGS has been instrumental in exploring clonal evolution and subclonal dynamics in AML, shedding light on the mechanisms of disease progression and treatment resistance [17]. WGS has also expanded our understanding of clonal evolution in AML by identifying subclonal populations that drive relapse after treatment. The identification of mutational signatures caused by environmental exposures and aging has also been pivotal in understanding AML progression. Papaemmanuil et al. (2016) linked specific mutations to prognosis, revealing the impact of these factors on disease evolution [3]. Alexandrov et al. (2020) expanded this by cataloging mutational signatures in cancers, showing that aging and environmental factors contribute significantly to AML’s genetic heterogeneity, influencing treatment responses and outcomes [24]. However, WGS’s clinical adoption is hindered by high costs and the complexity of data interpretation, requiring advanced bioinformatics infrastructure to analyze large datasets.

#### Third-Generation Sequencing Approaches

Third-generation sequencing platforms, such as Oxford Nanopore and PacBio, have revolutionized WGS by providing long-read sequencing capabilities that complement traditional short-read methods [25]. These technologies enable the sequencing of large genomic regions without fragmentation, which is advantageous for detecting complex structural variants like balanced translocations and inversions that are significant in AML. Oxford Nanopore excels in real-time sequencing and portability, while PacBio offers high-fidelity (HiFi) reads for exceptional accuracy [26,27]. Despite their potential, third-generation sequencing has challenges, including higher costs, specialized equipment requirements, and complex bioinformatics analysis. Nevertheless, these technologies have been successfully employed in AML research, yielding novel genetic insights and emphasizing their promise for clinical applications.

### 2.2. Whole-Exome Sequencing (WES)

Whole-exome sequencing (WES) focuses on sequencing the exonic, or protein-coding, regions of the genome, representing only 1–2% of the genome but containing 85% of known disease-causing mutations [28]. WES has been widely adopted in clinical diagnostics and research due to its cost-effectiveness compared to WGS. In AML research, WES has been crucial in identifying mutations in key genes involved in transcription regulation and epigenetic modification, such as CEBPA, RUNX1, and TP53 [2,3,29].

Recent studies have demonstrated the clinical relevance of WES in AML, particularly in identifying rare mutations that guide treatment decisions. In 2016, Papaemmanuil et al. introduced a genomic classification linking mutations to prognosis, revealing both known and novel mutations that influence disease progression [3]. De Kouchkovsky and Abdul-Hay (2016) highlighted WES’s role in detecting poor-prognosis mutations like FLT3 and NPM1 and tracking genetic changes to monitor treatment response and relapse [30]. Guijarro et al. (2024) further expanded WES’s application to intermediate-risk AML, uncovering hidden mutations and refining diagnostic categories for more tailored treatment [31]. Though WES is widely used, it does not capture large structural variants, a limitation that hinders comprehensive understanding in some AML cases. Recent research continues to explore ways to bridge this gap and enhance the resolution of WES.

### 2.3. Targeted Sequencing

Targeted sequencing, also known as gene panel sequencing, focuses on sequencing specific genomic regions or genes of interest, allowing for deep coverage and high sensitivity in detecting rare variants. This approach has become particularly useful in cancer genomics, where it is employed to identify clinically relevant mutations in oncogenes and tumor suppressor genes [32]. In AML, targeted sequencing panels typically include genes frequently mutated, such as FLT3, NPM1, DNMT3A, IDH1/2, and TP53 [3,33].

Recent advancements in targeted sequencing have further improved its accuracy and clinical utility in AML. A 2023 study highlighted the efficacy of targeted sequencing in guiding the use of FLT3 inhibitors in FLT3-mutated AML, significantly improving patient outcomes [34]. For example, quizartinib, a potent selective type 2 FLT3 inhibitor combined with chemotherapy, demonstrated antitumor activity and an acceptable safety profile in patients who are FLT3-ITD-positive and newly diagnosed with AML [35]. Although targeted sequencing excels in sensitivity, it can miss mutations in unexplored genomic regions, posing a limitation. Nevertheless, it remains a cornerstone for precision medicine in AML treatment.

## 3. Advances in RNA Sequencing

RNA sequencing (RNA-Seq) technologies have undergone significant advancements in recent years, enabling more comprehensive profiling of gene expression, splicing events, and non-coding RNAs. These technologies have become indispensable in the study of transcriptomics—the complete set of RNA transcripts expressed by a genome [36,37]. RNA-Seq allows researchers to unravel the complexity of gene expression and regulation, leading to the discovery of key biomarkers and therapeutic targets in various diseases, including cancer [28,37,38].

### 3.1. Transcriptome Sequencing Subtypes

RNA-Seq can be divided into multiple subtypes, each offering unique insights into gene expression. Messenger RNA (mRNA) sequencing, the most common subtype, focuses on identifying transcripts that carry genetic instructions from DNA to protein synthesis machinery [39]. Beyond mRNA sequencing, long non-coding RNA (lncRNA) sequencing and microRNA sequencing have gained prominence for their roles in gene regulation. Long non-coding RNAs have been found to participate in critical cellular processes, including epigenetic regulation and transcriptional control [40,41].

In cancer research, transcriptome sequencing has proven invaluable in identifying differentially expressed genes that serve as biomarkers for diagnosis and treatment. For instance, RNA-Seq has identified overexpressed lncRNAs in various cancers, correlating with poor prognosis [42,43]. Similarly, in AML, RNA-Seq has uncovered novel gene expression patterns and fusion transcripts, which have significant therapeutic implications, aiding in the development of targeted therapies [44,45]. For example, Kerbs et al. (2022) showed that RNA-Seq can detect fusion genes such as NRIP1-MIR99AHG, which play a key role in diagnosis and the advancement of targeted treatment options [44]. However, further advancements in RNA-Seq technology have allowed for the identification of novel gene expression patterns linked to therapeutic resistance, aiding in the development of targeted therapies.

### 3.2. Single-Cell RNA Sequencing

Single-cell RNA sequencing (scRNA-Seq) has revolutionized our understanding of cellular heterogeneity in tissues. By profiling gene expression at the level of individual cells, scRNA-Seq can uncover rare cell populations and distinct cellular states critical for disease progression [46,47]. This technique has been particularly impactful in cancer research, where it has revealed distinct subpopulations of cancer cells with varying levels of drug resistance [48,49].

In AML, scRNA-Seq has been instrumental in uncovering novel subpopulations of leukemia stem cells, which may contribute to drug resistance and relapse. Studies by van Galen et al. (2019) and Zhou & Chng (2024) revealed how scRNA-Seq can map the hierarchical structures within AML, uncovering distinct cellular states linked to disease progression and immune responses [13,14]. Furthermore, high-throughput single-cell genomics has highlighted the clonal evolution of AML, showing how resistant subpopulations can impact therapeutic outcomes. Research from Morita et al. (2020) and Velten et al. (2021) used clonal tracking to trace leukemic and pre-leukemic stem cell lineages, identifying potential targets for immunotherapy and other treatments aimed at eliminating these resistant populations [50,51]. These findings have opened new avenues for therapeutic intervention, providing potential targets for immunotherapy and other treatment strategies aimed at eradicating resistant cell populations. The application of scRNA-Seq continues to refine treatment strategies by illuminating cellular states critical to disease progression.

## 4. Clinical Applications

The clinical applications of DNA and RNA sequencing have expanded dramatically in recent years, providing new opportunities for diagnosing diseases, stratifying patients by risk, and personalizing therapies based on genetic and molecular profiles. These advancements have paved the way for precision medicine, where treatment decisions are tailored to the individual patient’s genetic makeup. Notably, recent studies emphasize the role of comprehensive genomic profiling in improving patient outcomes.

### 4.1. Diagnosis and Classification

Next-generation sequencing (NGS) technologies, including whole-genome sequencing (WGS) and whole-exome sequencing (WES), have become indispensable in diagnosing genetic disorders and cancers such as AML. These technologies allow for the precise classification of diseases based on molecular profiles, which in turn informs treatment strategies. WGS and WES have been particularly impactful in uncovering genetic conditions after years of inconclusive testing, leading to more accurate diagnoses [52,53].

In oncology, molecular profiling using these tools has revolutionized cancer classification, allowing for treatment based on tumor-specific molecular characteristics. According to Akhoundova and Rubin (2022), advanced multi-omics tumor profiling combines various approaches, including genomics, transcriptomics, and proteomics, to offer a comprehensive view of cancer biology, thus shaping the future of precision oncology [54]. For AML, molecular profiling aids in detecting critical mutations such as IDH1/2 and FLT3, which are essential for guiding personalized treatment regimens. Malone et al. (2020) noted that recognizing these molecular changes improves the selection of targeted therapies, enhancing patient outcomes [55]. Additionally, targeted RNA sequencing allows for the quantification of measurable residual disease, improving monitoring and enabling early detection of relapses in AML [56].

### 4.2. Prognosis and Risk Stratification

Genomic profiling is becoming a cornerstone for predicting disease outcomes and stratifying patients by risk, particularly in cancer. In breast cancer, tools such as the RNA-Seq-based Oncotype DX test help predict recurrence risk and guide chemotherapy decisions [57,58]. This molecular assay measures the expression of specific genes associated with breast cancer prognosis, enabling oncologists to tailor treatment plans to individual patients effectively. In AML, the identification of specific mutations, such as FLT3-ITD, TP53, and NPM1, plays a crucial role in stratifying patients’ risk profiles, directly influencing prognosis and treatment strategies [59,60]. For instance, the presence of FLT3-ITD mutations is linked to a more aggressive disease course, while mutations in TP53 may indicate poor prognosis. This genetic information assists clinicians in determining whether a patient is a candidate for allogeneic stem cell transplantation, a critical consideration that can significantly impact patient survival.

Recent studies emphasize the importance of integrating clinical and molecular data for enhanced risk stratification in AML. The presence of additional mutations alongside FLT3-ITD can further refine prognostic assessments and inform treatment recommendations. For example, co-occurring mutations may alter the efficacy of certain therapies, necessitating a more tailored approach to patient management [61]. This integrative strategy not only facilitates a more individualized treatment approach but also optimizes therapeutic outcomes through informed clinical decision-making.

### 4.3. Targeted Therapy and Personalized Medicine

One of the most exciting clinical applications of NGS is the development of targeted therapies, which have transformed cancer treatment. NGS technologies enable the identification of specific mutations driving disease, allowing clinicians to prescribe therapies tailored to individual genetic profiles. For instance, PARP inhibitors have been developed for patients with BRCA1/2 mutations in breast and ovarian cancers, significantly improving outcomes [62,63]. Similarly, ALK inhibitors are now commonly used to treat ALK-positive non-small cell lung cancer (NSCLC) based on genetic profiling [64,65].

In AML, RNA-Seq has gained traction in personalized medicine. RNA-Seq helps identify fusion genes and aberrant gene expression patterns, allowing for the development of targeted therapies. For example, RNA-Seq has been instrumental in identifying fusion transcripts, such as KMT2A rearrangements, which can be therapeutically targeted with small molecule inhibitors [66,67]. More recently, the discovery of novel biomarkers through transcriptomic profiling has enabled the use of epigenetic modifiers and small molecule inhibitors to target specific pathways involved in leukemogenesis, providing new treatment avenues for patients with otherwise poor prognoses [68,69,70].

### 4.4. DNA vs. RNA Sequencing

While DNA and RNA sequencing provide critical insights into diseases’ genetic and molecular basis, they offer complementary information. DNA sequencing gives a static view of the genome, identifying genetic mutations, structural variants, and copy number changes that are stable across an individual’s lifetime. This makes it ideal for identifying hereditary mutations and cancer-related oncogenic drivers, such as KRAS or TP53, which are pivotal in tumor initiation and progression [28,71].

On the other hand, RNA sequencing (RNA-Seq) offers a dynamic view by analyzing gene expression in real time, revealing how genes are transcribed and translated under different physiological or pathological conditions [72,73]. RNA-Seq is particularly effective in detecting abnormal gene expression patterns, such as alternative splicing events, that contribute to disease progression and therapeutic resistance, especially in cancers [71]. Recent research emphasizes how transcriptomic changes can inform the discovery of new therapeutic targets in AML, further bridging the gap between genomic insights and clinical applications [74].

The integration of both DNA and RNA sequencing provides a more comprehensive understanding of disease mechanisms, which in turn leads to more effective therapeutic interventions and improved patient outcomes [73,75]. DNA sequencing uncovers the genetic mutations that predispose individuals to diseases, while RNA-Seq captures the downstream effects of those mutations, such as aberrant gene expression, allowing for a more detailed assessment of the molecular pathways involved in disease progression. This combined approach has proven invaluable in cancer research, where understanding both the genetic and transcriptional changes in tumor cells can lead to more targeted and effective treatments, enhancing patient outcomes [28,76,77].

By using both sequencing methods, researchers and clinicians can identify genetic mutations through DNA analysis and assess their functional impacts via RNA expression studies. This multi-omics approach significantly improves diagnostic accuracy and therapeutic interventions, enriching our understanding of intricate diseases, particularly AML. The continuous evolution of these technologies promises to further refine patient stratification and treatment personalization in the near future.

## 5. Challenges and Future Directions

The landscape of DNA and RNA sequencing in acute myeloid leukemia (AML) is evolving rapidly, but numerous challenges persist. Addressing these challenges is crucial for advancing clinical practice and improving patient outcomes.

### 5.1. Data Complexity and Interpretation

Next-generation sequencing (NGS) technologies generate vast amounts of data, necessitating complex computational tools for data storage, analysis, and interpretation. The sheer volume of data and the challenge of distinguishing between benign variants and pathogenic mutations remain key hurdles. Variants of uncertain significance (VUSs) often complicate clinical decision-making, particularly in the context of whole-genome sequencing (WGS) [78]. To tackle these challenges, researchers are developing more advanced bioinformatics tools and machine-learning algorithms aimed at enhancing variant interpretation. Recent advancements in deep learning and artificial intelligence have significantly enhanced the accuracy of predicting the pathogenicity of novel mutations [79]. Additionally, open-access genomic databases, such as gnomAD and ClinVar, are expanding, providing clinicians with more comprehensive resources for interpreting genomic data [80].

Another interesting development is the integration of multi-omic approaches that combine genomic, transcriptomic, and epigenomic data, which can provide a more comprehensive understanding of AML biology and improve prognostic models. This holistic view is crucial for developing targeted therapies and improving patient stratification.

### 5.2. Cost and Accessibility

Although the costs of sequencing have decreased significantly over the past decade, both whole-genome sequencing (WGS) and whole-exome sequencing (WES) continue to be prohibitively expensive for widespread routine clinical application, especially in resource-limited settings [81]. The financial burden encompasses not only sequencing but also data storage, interpretation, and bioinformatics, posing significant obstacles for healthcare systems worldwide.

Innovations such as cloud-based platforms for genomic analysis have been developed to reduce infrastructure costs and democratize access to sequencing data, facilitating large-scale genomic studies [82,83]. However, in clinical practice, the integration of these platforms remains limited by institutional constraints and a lack of standardized workflows for data interpretation. This limitation is especially prominent in low-resource environments where high-throughput technologies are not widely accessible, despite ongoing technological advancements. Portable sequencing technologies, such as third-generation platforms from Oxford Nanopore and PacBio, present promising solutions for rapid, on-site genomic analysis [84]. These platforms offer long-read sequencing with real-time capabilities, providing more accurate genomic mapping while reducing the need for costly, labor-intensive methods. Despite challenges like higher upfront costs and specialized equipment, these technologies have the potential to improve patient access to diagnostics and treatment, particularly in underserved regions, thus enabling early diagnosis and personalized treatment strategies for diseases like AML.

### 5.3. Ethical and Privacy Concerns

As next-generation sequencing (NGS) technologies become more prevalent in healthcare, ethical and privacy concerns have gained increasing attention. A significant issue is the potential misuse of genetic information, especially in employment and insurance contexts, where it could precipitate discrimination. Ensuring the privacy and security of genetic data has become paramount, and advancements in encryption and anonymization technologies aim to mitigate these risks.

Additionally, the widespread use of genetic data in research, particularly in large-scale genomic studies, raises ethical concerns about informed consent and data sharing across international borders. These issues are amplified by the complexities of managing consent in studies involving multiple institutions, with varying regulations and ethical standards [85]. To address these concerns, new guidelines are being established, focusing on improving transparency, strengthening data sharing regulations, and ensuring more robust patient education about the implications of genomic testing [85].

In response to these challenges, international organizations are developing stronger ethical frameworks and legal protections to ensure that genetic data is used responsibly while minimizing risks to individual privacy [85]. This includes efforts to create standardized protocols for informed consent and the secure sharing of genomic information, which is vital for the continued growth of precision medicine.

## 6. Conclusions

The rapid evolution of DNA and RNA sequencing technologies has fundamentally transformed our understanding of human biology and the genetic underpinnings of various diseases. Innovations such as whole-genome sequencing (WGS) and whole-exome sequencing (WES) have significantly advanced genetic diagnostics, enabling the identification of both common and rare mutations across a spectrum of conditions (Table 1). Additionally, RNA sequencing (RNA-Seq) has provided critical insights into gene expression, uncovering dynamic transcriptional changes that are essential for elucidating complex disease mechanisms, particularly in cancer research.

In the realm of acute myeloid leukemia (AML), these technologies have proven instrumental in identifying pivotal genetic mutations and transcriptional alterations that drive disease progression. WGS provides detailed insights into structural changes, clonal populations, and relapse-associated variants, while third-generation platforms like Oxford Nanopore and PacBio enhance precision with long-read capabilities. Moreover, the integration of multi-omic approaches combining genomic, transcriptomic, and epigenomic data has emerged as a particularly promising avenue, allowing for a more comprehensive understanding of AML biology. This holistic strategy enhances classification, risk stratification, and the development of targeted therapies tailored to individual genetic profiles. Furthermore, recent advancements in artificial intelligence and machine learning are refining the analysis of complex data sets, enabling better prediction of treatment responses and the identification of novel therapeutic targets. While challenges related to data complexity, costs, and ethical considerations remain, continuous innovation in sequencing technologies holds great potential for revolutionizing AML diagnosis and treatment, promoting a future where personalized and effective therapies substantially improve patient outcomes.

## Figures and Tables

**Table 1 ijms-26-00071-t001:** Advances in DNA and RNA sequencing technologies and their benefits for patients with acute myeloid leukemia (AML).

Sequencing Technology	Description	Advances and Benefits in AML	References
Whole-Genome Sequencing (WGS)	Comprehensive sequencing of the entire genome, including both coding and non-coding regions, at once.	– Identifies mutations and structural variations in AML; – Aids understanding of clonal evolution and treatment resistance; – Enables targeted therapies by revealing genetic alterations; – Treatment decision insights.	[16,86,87,88,89,90,91,92]
DNA Breakpoint Cloning with WGS	A technique for detecting minimal residual disease (MRD) using WGS for a detailed analysis of genetic variations, including structural changes.	– Enhances MRD detection with superior sensitivity compared to PCR; – Identifies clonal populations and variants linked to relapse.	[19,20]
Third-Generation Sequencing (Oxford Nanopore, PacBio)	Platforms providing long-read sequencing capabilities for analyzing large genomic regions without fragmentation.	– Detects complex variants like balanced translocations and inversions; – Oxford Nanopore: real-time sequencing and portability; – PacBio: high-fidelity (HiFi) reads for accuracy.	[25,26,27]
Whole-Exome Sequencing (WES)	Sequencing of all protein-coding regions of the genome to identify genetic variants associated with diseases.	– Detects mutations in key AML-related genes (CEBPA, RUNX1, and TP53) involved in transcription regulation; – Aids in diagnosing AML subtypes based on specific genetic markers, guiding personalized treatments; – Identifies actionable mutations in AML to guide targeted therapies and improve patient outcomes.	[93,94,95,96,97,98,99]
Targeted Sequencing	Focuses on specific genes or genomic regions associated with diseases like AML, targeting known genetic drivers.	– Detects rare mutations in AML-related genes (e.g., FLT3, NPM1, DNMT3A, IDH1/2, and TP53); – Guides targeted therapies (e.g., FLT3 inhibitors) to improve outcomes; – Improves sensitivity for mutations in specific pathways relevant to AML, facilitating better disease understanding and management.	[56,100,101,102,103,104,105,106]
RNA Sequencing (RNA-Seq)	Sequencing of RNA transcripts to determine gene expression profiles. Provides a snapshot of active/inactive genes in different conditions by sequencing all RNA in a sample.	– Reveals gene expression patterns and identifies fusion transcripts with therapeutic relevance; – Assists in developing targeted therapies by uncovering abnormal gene expression profiles; – Assists in the development of targeted therapies by revealing aberrant gene expression profiles specific to AML; – Enhances understanding of the AML microenvironment and treatment resistance pathways, revealing potential novel therapeutic targets.	[107,108,109,110,111]
Single-Cell RNA Sequencing (scRNA-Seq)	Sequencing RNA from individual cells to assess cellular heterogeneity. Profiles the transcriptome at the single-cell level, revealing cellular diversity and rare cell types within tumors.	– Identifies rare subpopulations of leukemia stem cells contributing to disease progression and treatment failure; – Offers new targets for immunotherapy and strategies to eliminate resistant cells; – Provides insights into AML heterogeneity, enabling personalized therapeutic strategies.	[112,113,114,115]

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
