# Peer review of "Advances in DNA/RNA Sequencing and Their Applications in Acute Myeloid Leukemia (AML)"

_ijms, 2024, doi:10.3390/ijms26010071_

Round 1

Reviewer 1 Report

Comments and Suggestions for Authors

The authors proposed a deep review on all the strategies based on  NGS approaches for the diagnosis, prognosis and possible personalized approaches in AML patients. The review is well written and most of the crucial points have been discussed.

There are some points I suggest to improve:

 -          In section 2.1: I suggest to better specify that among the structural variations, the balanced translocations and inversions are frequently associated to AML and WGS could easily detect the translocations and inversions with implication for the diagnosis and comparison with standard approaches (i.e. FISH, karyotype) might be proposed.

-          In the methodology section, the third generation sequencing paltforms (such as  Oxford Nanopore and PacBio) is lacking. I suggest to add a subpoint of the 2.1 with the new long reads WGS approaches with pros and cons to these technologies. For instances, some authors used third generation sequencing in AML. Costs might be discussed in point 5.2

-          The DNA breakpoint cloning approaches for the minimal residual disease monitoring is lacking. Many groups started publishing firstly on CML breakpoits with related minimal residual disease and personalized medicine implications. Recently, also on AML, NGS breakpoint cloning approaches were implemented. I suggest to add a dedicated paragraph to present this approach with related literature. A discussion of this approach should also be added to conclusions.

-          Table 1 shoud be modified according to the observation of the above points

Author Response

Comment 1: In section 2.1: I suggest to better specify that among the structural variations, the balanced translocations and inversions are frequently associated to AML and WGS could easily detect the translocations and inversions with implication for the diagnosis and comparison with standard approaches (i.e. FISH, karyotype) might be proposed.

Response 1: Thank you for the suggestion. We have incorporated this suggestion and specified the role of balanced translocations and inversions in AML detection via WGS. The revised content can be found on page 2, lines 74-83.

Comment 2: In the methodology section, the third generation sequencing paltforms (such as  Oxford Nanopore and PacBio) is lacking. I suggest to add a subpoint of the 2.1 with the new long reads WGS approaches with pros and cons to these technologies. For instances, some authors used third generation sequencing in AML. Costs might be discussed in point 5.2

Response 2: As suggested, we have added a section discussing third-generation sequencing technologies and their implications for AML. The revised content is found on page 3, lines 99-110; page 7, lines 327-334; and page 9, lines 372-375.

Comment 3: Add a paragraph on DNA breakpoint cloning approaches for minimal residual disease (MRD) monitoring in AML, including related literature.

Response 3: We have added a dedicated paragraph on this topic, addressing the role of DNA breakpoint cloning approaches in AML MRD monitoring. The revised content is found on page 2, lines 64-73.

Comment 4: Modify Table 1 according to the feedback.
Response 4: We have updated Table 1 to reflect the revisions suggested by the reviewer.

Reviewer 2 Report

Comments and Suggestions for Authors

This review is interesting but limited for my opinion.

Most paragraphs are limited and have to be improved, describing more in depth the results obtained in the works reported as references.

After that, it might be suitable for publication.

For example: 

The identification of mutational signatures caused by environmental exposures and aging has also been pivotal in understanding AML progression.[3,15] However, WGS’s clinical adoption is hindered by high costs and the complexity of data interpretation, requiring advanced bioinformatics infrastructure to analyze large datasets.

DESCRIBE MORE IN DEPTH

Recent advancements in targeted sequencing have further improved its accuracy and clinical utility in AML. A 2023 study highlighted the efficacy of targeted sequencing in guiding the use of FLT3 inhibitors in FLT3-mutated AML, significantly improving patient outcomes.

DESCRIBE MORE IN DEPTH

and there are more of these sentences

Author Response

Comment: This review is interesting but limited for my opinion.

Most paragraphs are limited and have to be improved, describing more in depth the results obtained in the works reported as references.

After that, it might be suitable for publication.

For example: 

The identification of mutational signatures caused by environmental exposures and aging has also been pivotal in understanding AML progression.[3,15] However, WGS’s clinical adoption is hindered by high costs and the complexity of data interpretation, requiring advanced bioinformatics infrastructure to analyze large datasets.

DESCRIBE MORE IN DEPTH

Recent advancements in targeted sequencing have further improved its accuracy and clinical utility in AML. A 2023 study highlighted the efficacy of targeted sequencing in guiding the use of FLT3 inhibitors in FLT3-mutated AML, significantly improving patient outcomes.

DESCRIBE MORE IN DEPTH

and there are more of these sentences

Response: Thank you for the suggestion. We have expanded on the results reported in the referenced works, providing more in-depth descriptions as requested. The revised content is located on page 2, lines 92-96; page 3, lines 119-125 and lines 141-143; page 4, lines 167-169 and lines 181-189; and page 5, lines 210-218 and lines 225-244.

Reviewer 3 Report

Comments and Suggestions for Authors

In the manuscript submitted to me for review entitled "Advances in DNA/RNA Sequencing and Their Applications in Acute Myeloid Leukemia (AML) the authors Fatimah Ahmed and Jiang Zhong present a study on various DNA and RNA sequencing technologies, including whole genome sequencing (WGS), whole exome sequencing (WES) and RNA sequencing (RNA-Seq), which are used to better understand Acute Myeloid Leukemia (AML) at the molecular level. The systematization of the available information can lead to the improvement of the various techniques and their real application in practice, in order to improve the diagnosis of the disease and to develop personalized therapy, thereby improving the prognosis for patients.

The review presents studies by many authors presenting different sequencing techniques. 101 references covering data mostly from the last fifteen years are included to support the study. About 2/3 of the references are from the last 5 years, which shows that the topic is relatively new and the interest in better understanding of the mechanism of cancer occurrence through the use of different sequencing methods is constantly increasing. This suggests that the present manuscript would attract the attention of IJMS readers. I did not notice any redundant self-citations, all references used are appropriate and necessary for the preparation of the manuscript. The research presented in this manuscript has been carried out in an extremely thorough manner. That is why I do not have any significant remarks, I have only two recommendations for the authors.

1. If more data about AML can be presented in the first paragraph of the introduction. If the authors find, for example, some more statistics about the disease: for example, % of patients diagnosed in a year or some time period; in what % of patients the therapy is effective; in what % of patients relapses occur, etc.

2. Almost all references in the "References" section are presented with only the first three authors and "et al". As required by MDPI "et al." is placed with more than 10 authors. I think it will be helpful for the reader to add the missing authors. See instructions for authors.

·       For documents co-authored by a large number of persons (more than 10 authors), you can cite the first ten authors, then add a semicolon and add 'et al.' at the end:

Author 1; Author 2; Author 3; Author 4; Author 5; Author 6; Author 7; Author 8; Author 9; Author 10; et al.

Author Response

Response to Reviewer #3:

Comment 1: If more data about AML can be presented in the first paragraph of the introduction. If the authors find, for example, some more statistics about the disease: for example, % of patients diagnosed in a year or some time period; in what % of patients the therapy is effective; in what % of patients relapses occur, etc.

Response 1: Thank you for the suggestion. AML statistics have been added to the first paragraph of the introduction, now on page 1, lines 27-33.

Comment 2: Almost all references in the "References" section are presented with only the first three authors and "et al". As required by MDPI "et al." is placed with more than 10 authors. I think it will be helpful for the reader to add the missing authors. See instructions for authors.

  • For documents co-authored by a large number of persons (more than 10 authors), you can cite the first ten authors, then add a semicolon and add 'et al.' at the end:

Author 1; Author 2; Author 3; Author 4; Author 5; Author 6; Author 7; Author 8; Author 9; Author 10; et al.

Response 2: As suggested, we have updated all references with more than 10 authors to list the first ten authors followed by "et al."

Round 2

Reviewer 2 Report

Comments and Suggestions for Authors

Thanm you for your revisions, now the work is improved and suitable for publication